# In Vitro Transfection of Up-Regulated Genes Identified in Favorable-Outcome Neuroblastoma into Cell Lines

**DOI:** 10.3390/cells11193171

**Published:** 2022-10-10

**Authors:** Yoko Hiyama, Emi Yamaoka, Takahiro Fukazawa, Masato Kojima, Yusuke Sotomaru, Eiso Hiyama

**Affiliations:** 1Biomedical Science Division, Natural Science Center for Basic Research and Development (N-BARD), Hiroshima University, 1-2-3, Kasumi, Minami-ku, Hiroshima 734-8551, Japan; 2Department of Pediatric Surgery, Hiroshima University Hospital, Hiroshima University, 1-2-3, Kasumi, Minami-ku, Hiroshima 734-8551, Japan

**Keywords:** neuroblastoma, cell line, *DHRS3*, *NROB1*, *CYP26A1*, transfection, siRNA, ATRA, senescence, differentiation

## Abstract

We previously used microarrays to show that high expression of *DHRS3*, *NROB1*, and *CYP26A1* predicts favorable NB outcomes. Here, we investigated whether expression of these genes was associated with suppression of NB cell (SK-N-SH, NB12, and TGW) growth. We assessed morphology and performed growth, colony-formation, and migration assays, as well as RNA sequencing. The effects of the transient expression of these genes were also assessed with a tetracycline-controlled expression (Tet-On) system. Gene overexpression reduced cell growth and induced morphological senescence. Gene-expression analysis identified pathways involving cellular senescence and cell adhesion. In these cells, transduced gene dropout occurred during passage, making long-term stable gene transfer difficult. Tet-On-induced gene expression caused more pronounced cell-morphology changes. Specifically, *DHRS3* and *NROB1* led to rapid inhibition and arrest of cell growth, though *CYP26A1* did not affect cell-growth rate or cell cycle. *DHRS3* arrested the cell cycle by interacting with the all-trans-retinol pathway and drove differentiation and senescence in tumors. Overexpression of these genes reduced the malignant grade of these cells. A new therapeutic strategy might be the induction of these genes, as they suppress the growth of high-risk neuroblastoma and lead to differentiation and senescence.

## 1. Introduction

Neuroblastoma (NB) is one of the most common solid tumors occurring in children. This tumor is divided into favorable and unfavorable biological categories, with the unfavorable tumors being more common in older children and characterized by rapid progress and chemo-resistance. Favorable tumors usually occur in infants, and some even regress or mature spontaneously. Since the malignant grade of NB biology greatly influences patient outcomes, biological factors related to the malignant grade of NB are of great clinical interest but are not yet fully understood.

In our department, we performed an exhaustive genetic analysis of biologically favorable and unfavorable NB tumors derived from 16 favorable cases who were alive with remaining residual tumors, including 4 spontaneous regressed-stage MS cases, and from clinical 16 unfavorable cases who died of tumor progression, and extracted genes with significantly higher expression levels in favorable tumors with oligo-microarrays [1,2,3]. The top three highly expressed genes in favorable tumors were *DHRS3*, *NROB1* (also known as *DAX1*), and *CYP26A1*, which are correlated with functions involving differentiation of the nervous system and cell division arrest [4,5,6]. These genes are activated in neuronal development and differentiation. Therefore, expression levels of these genes might mediate the regression or maturation of NB cells.

NB cell lines were established from an aggressive neuroblastoma case (Appendix A). In this study, we transfected each of these three genes into NB cell lines (SK-N-SH, NH12, and TGW, which have different characteristics) using the plasmid method of expression. To knockdown these genes, siRNA for each gene was also transfected into these cells. Subsequently, cell morphology was assessed, and assays for growth, cell cycle, migration, and colony formation to monitor anchorage-independent growth were performed. Gene expression profiles were further examined by RNA sequencing. Transfection of these highly expressed genes reduced cell growth, which consequently made it difficult to establish a stable cell-induced differentiation assay. To rule out the effect of the transfection process of these genes, we also used the retroviral gene stable expression system known as the Retro-X Tet-On Advanced System. To assess the correlation with NB differentiation induced by all-trans retinoic acid (ATRA), cells in which the Tet-On expression system was used were selected.

## 2. Materials and Methods

### 2.1. Cell Culture

TGW and SK-N-SH cells were obtained from the Japanese Cancer Resource Cell Bank (http://cellbank.nihs.go.jp/, accessed on 7 March 2003) and the NH-12 cell line was established in our institution. TGW cells were cultured in Dulbecco’s Modified Eagle Medium, and SK-N-SH and NH12 cells were cultured in MEM-alpha (GIBCO BRL, Rockville, MD, USA) containing 2 mM L-glutamine. These media were supplemented with 10% fetal bovine serum (FBS) (Labome, Princeton, NJ, USA) and 100 µg/mL penicillin/streptomycin. Passage numbers of these cell lines were largely equal.

### 2.2. Plasmid Construction and Recombinant Retrovirus

To generate recombinant plasmid constructs, we amplified full-length *Homo sapiens* dehydrogenase/reductase 3 (DHRS3, RefSeqID, NM_004753, transcription variant 1) cDNA, *Homo sapiens* cytochrome P450 family 26 subfamily A member 1 (CYP26A1, RefSeqID, NM_000783 transcription variant 1) cDNA, and *Homo sapiens* nuclear receptor subfamily 0 group B member 1 (NROB1, RefSeqID, NM_000475) cDNA by PCR, using the following primer sets:

DHRS3-F: 5′- ATATGGTGTGGAAACGGCTG-3′, DHRS3-R: 5′-ATCCTATGTCCGCCCTTTGAA-3′; CYP26A1-F: 5′- ATATGGGGCTCCCGGCGCT-3′, CYP26A1-R: 5′- TCAGATTTCCCCATGGAAATGGGT-3′; NROB1-F: 5′-ATATGGCGGGCGAGAACCACCA-3′, NROB1-R: 5′- TTATATCTTTGTACAGAGCATTTCCAGCA-3′. These PCR products were cloned and inserted into a pIRES2 DsRed-Express2 vector (Clontech Lab, Takara bio, Mountain View, CA, USA), and then a 3xFLAG tag was inserted at the N-terminal of each gene.

These products were also used in the Retro-X™ Tet-*On*^®^ Advanced Inducible Expression System (Clontech) which was applied to generate retrovirus constructs. Then, we established stable expression of each retroviral candidate gene for NB by expressing genes that activate transcription of downstream genes in the presence of doxycycline (Dox). We then confirmed expression following induction with Dox and examined changes following induction. The HEK-293-based retroviral packaging cell line GP2-293 was used for packaging retroviruses. Induction of gene expression was achieved by adding Dox (Clontech). Sequences were checked using the PRISM 3130 sequencer (ABI, Thermo Fisher Scientific, Waltham, MA, USA).

The day before transfection, culture cells were trypsinized and counted. The cells were seeded in a 24-well plate to achieve 50~70% confluency on the day of transfection in 100 µL of Serum free Opti-MEM^®^ I Reduced Serum Medium with 500 ng of plasmid DNA and then 0.5 μL of PLUS Reagent was added. After incubation for 5 min at room temperature (r.t.), the cells were mixed thoroughly with Lipofectamine LTX and then incubated for 30 min at r.t. These cells were divided into each well and incubated for 6 hrs at 37 °C with 5% CO2. The selection process using G418 and Blasticidin, as described below, started from the next day. The selected clones were confirmed for expression of transfected genes using Western blotting and real-time quantitative RT-PCR (qRT-PCR), as described below.

### 2.3. In Vitro Cell-Migration Assay

An in vitro cell-migration assay was performed using a culture insert (ibidi, Gewerbehof, Germany). Cells (2 × 10^5^) were seeded in the two wells of a culture insert, and after the cells adhered, the insert for producing the gap was removed. After 0 h, 24 h, and 48 h, the migration of cells into the gap area was analyzed by ImageJ software [7].

### 2.4. Soft-Agar Colony-Formation Assay

We subjected cells to the soft-agar colony-formation assay in 35 mm dishes; 2000 cells were seeded into 1.5 mL of pre-autoclaved 0.8% low-melting agarose in filtrated 2× DMEM or a-Medium supplied with 20% FBS, which was overlaid with a 1.5 mL layer of 0.4% agar in the 2× appropriate medium. After 3 weeks, the colonies were analyzed and counted, and, after fixation, they were stained with crystal violet. Image-Pro software (Media Cybernetics, Rockville, MD, USA) was used for this assay.

### 2.5. Cell-Proliferation Assay

Cells were seeded (2 × 10^5^) into 35 mm dishes and cultured in DMEM/alpha-MEM supplemented with 10% FBS and then harvested and counted using the LUNA automatic-cell-counting system (Logos Biosystems, Annandale, VA, USA) every two days. Proliferation was assessed in triplicate, and each experiment was repeated three times.

### 2.6. Cell-Cycle Analysis

Cell cycles were analyzed using a flow cytometer (LSRFortessa X-20^TM^, Becton Dickinson Biosciences, Franklin Lakes, NJ, USA). Cells were harvested and rinsed with PBS and suspended in PBS with 1% bovine serum albumin (BSA). They were then stained with Hochest 33342. The staining solution was incubated for 30 min in the dark and the cells were then washed with PBS three times. Nuclear DNA was analyzed by flow cytometry, and the population of cells in each cycle was estimated.

### 2.7. Western Blot Analysis, Immunocytochemistry, and Electron Microscopy

Cells were harvested and whole-cell lysates were extracted using a reagent kit (cOmplete Lysis-M, Roche, Basel, Switzerland), according to the manufacturer’s protocol, and acetone protein precipitation was then performed. After centrifugation at 15,000× *g* for 10 min at 4 °C, debris was discarded, and protein concentrations were measured. SDS–polyacrylamide gel electrophoresis (12%) was then carried out. For these analyses, 20 μg of protein was added to each lane. Anti-FLAG M2 (Sigma, St. Louis, MO, USA) was used as the primary antibody, and horseradish-peroxidase-conjugated goat anti-mouse antibodies were used as the secondary antibodies (GE Healthcare, Chicago, IL, USA). Western blot signals were detected with the chemiluminescence substrate of the ECL™ Prime Western Blotting detection system (GE Healthcare). Immunohistochemistry was assessed as follows: NB cells were seeded on type-1-collagen-coated glass cover slips. After incubation, cells were fixed by 4% paraformaldehyde in PBS and permeabilized by 0.2% Triton X-100, and they were then washed with PBS, blocked by 10% normal goat serum (Thermo Fisher Scientific, Waltham, MA, USA), and incubated with the primary antibody mixture in 1% normal goat serum for 1 hr at room temperature. After the incubation and PBS washing steps, samples were covered with secondary antibody staining solution and incubated for 1 h at room temperature. Samples were then washed several times with PBS and visualized by fluorescence microscopy (BZ-X710, Keyence, Osaka, Japan), confocal microscopy (Fluoview 1000-D, Olympus, Tokyo, Japan), and super-resolution microscopy (DeltaVision OMX 3D-SIM, GE Healthcare, Houston, TX, USA). Expression of exogenous proteins was detected with FLAG M2 antibodies and their respective endogenous antibodies, as described below. Forced expression strains were also fat-stained using Lipid Tox. For the detection of endogenous protein expression, we used the following primary antibodies: mouse anti-DHRS3 (H00009249-B01P, Abnova, Taipei, Taiwan), rabbit anti-DHRS3 (NBP1-80846, NOVUS, Tokyo, Japan), mouse anti-CYP26A1 (sc-53618, Santa Cruz Biotechnology, Inc., Dallas, TX, USA), mouse anti-DAX1 (MABD398, Millipore, Burlington, MA, USA), and rabbit anti-DAX-1 (sc-841, Santa Cruz Biotechnology, Inc.). Mouse anti-β-actin (ab8226, Abcam, Cambridge, UK) and rabbit anti-β-actin (Abcam, ab8227) were also used to confirm loading amounts.

For the analysis of intracellular changes using electron microscopy, the culture cells were fixed with 3% glutaric dialdehyde/0.1 M PBS (pH 7.2), followed by 1% osmium tetraoxide (OsO4/0.1 M PBS), and stained with 3% uranyl acetate. The cells were absolutely dehydrated with an ascending series of ethanol solutions and embedded in Epon 812 resin. After confirming the resin cure, thin slices were cut in the vertical direction to form cell layers. The sections were laid on a copper grid and stained with 2% uranil acetate and lead citrate. Electron-microscopical examination was performed using a transmission electron microscope (JEM-1230, JEOL, Tokyo, Japan) under the conditions of an 80 kv accelerating voltage and a 20,000-magnification ratio.

### 2.8. Establishment of NB Cell Lines Expressing Favorable Candidate Factors

Transient and stable transfection in NB cell lines were assessed using the lipofection method (Lipofectamine LTX, Invitrogen, Thermo Fisher Scientific), according to the manufacturer’s protocol. After three weeks of selection with G418 and Blasticidin, the BD FACS Aria II (Becton Dickinson) flow cytometer was used to sort DsRed and EmGFP expression in NB cells. Isolated colonies were expanded and established as stable polyclonal or clonal clones.

### 2.9. Real-Time Quantitative RT-PCR (qRT-PCR)

qRT-PCR was performed using the Power Up SYBR™ Green RT qPCR kit (Thermo Fisher Scientific). RT-qPCR was run on the Bio-Rad CFX96 Real-Time PCR Detection System. The thermal cycling conditions were 50 °C for 2 min and 95 °C for 2 min, followed by 40 cycles at 95 °C for 1 s and 60 °C for 30 s. A melting-curve analysis was recorded after each cycle. Cycle thresholds (Cts) were calculated using Bio-Rad CFX Manager software. Relative gene expressions were calculated using the 2^−ΔΔCT^ method.

The primer sequences used were as follows (ACTB was used as the internal control): ACTB-F: 5′-TACCTCTATGCCAACACAGT-3′, ACTB-R: 5′- AGTACTTGCGCTCAGGAGGA-3′; DHRS3-F: 5′- TGGTCCATGGGAAGAGC-CTA-3′, DHRS3-R: 5′- TGTCCGCCCTTTGAAAGTGT-3′; CYP26A1-F: 5′-CTCCTCCAAATGGAATGAAG-3′, CYP26A1-R: 5′-CAGTCAGCATGGATGATATG-3′; NROB1-F: 5′- CAGTCAGCATGGATGATATG-3′, NROB1-R, 5′-CACAGCTCTTTATTCTTCCC-3′.

### 2.10. RNA Sequencing and Pathway Analysis

Comprehensive RNA-expression analysis was performed in DsRed- and EGFP-positive NB cells, as sorted by the FACS Aria cell sorter (Becton Dickinson Bioscience, Sparks, MD, USA). Total RNA was extracted from cell lines using the QIAwave RNA Mini Kit (Qiagen, Hilden, Germany). Total cellular RNA was extracted from tumor tissues by the acid-guanidinium–phenol–chloroform method [8]. The extracted RNA was quantified using a 2100 Bioanalyzer (Thermo Fisher Scientific, Santa Clara, CA, USA) with an Agilent RNA 6000 Nano kit.

RNA sequencing was performed as follows. Sequencing libraries were generated using the TruSeq RNA LT Sample Prep Kit (Illumina, SanDiego, CA, USA), according to the manufacturer’s instructions, with RNA normalized to inputs of 1 μg of total RNA from each sample. mRNA-sequence libraries were prepared from 500 ng of cDNA, according to the manufacturer’s protocol. Sequencing was performed for 101 cycles using the HiSeq Reagent Kit v3 on a HiSeq^®^ 2500 instrument (Illumina). The next-generation sequencing platforms for analysis and data visualization applied belonged to the CLC Genomics Workbench (Qiagen). After read genome annotation from Fast-q files, quality trimming, PCR-duplicate-mapped-reads removal, and filtering of reads, the transcripts were processed. The BAM file data were exported to Strand NGS v2.1 (Agilent Scientific Instruments, Santa Clara, CA, USA), and raw transcript counts were normalized by cell line with the parameters from DESeq2 software [9]. Then, expression analysis, gene ontology (GO) analysis, and pathway analysis were carried out. Ingenuity pathway analysis by IPA (TOMY digital biology, Tokyo, Japan) was applied for further investigation of biological pathway interactions.

### 2.11. Statistical Analysis

Unless otherwise specified, Fisher’s exact test was used for *p*-value calculations between two categorical variables. The Wilcoxon rank-sum and Kruskal–Wallis tests were used to analyze differences between two or more continuous variables. Log-rank tests were used to compare survival distributions in Kaplan–Meier plots. FDR corrections were performed for multiple testing corrections. Statistical analysis was carried out using the software programs RStudio, JMP v13, and Strand NGS. Statistical significance was defined at *p* < 0.05 by the Student’s *t*-test and Dunnett’s test. Results are presented as means ± SEs.

## 3. Results

### 3.1. Constitutive Transfection Analysis

The clones established after gene transfection were examined (Figure 1). Morphological changes representative of cellular senescence, such as enlargement and flattening of cells, were observed. We established that the overexpressed clones did express the exogenous proteins by confirming the expected molecular weights via Western blotting.

In the DHRS3-induced cell lines, protein expression levels of strains with a stably expressed transfected gene were assessed with fluorescent immunostaining (Figure 1A). Expression of exogenous proteins was detected with FLAG M2 antibodies. Among the transfected cell lines, DHRS3 was expressed mainly in the plasma membrane, endoplasmic reticulum (ER), and nucleus (Figure 1A(a,c)). The DHRS3-transfected strain was also observed by fat staining using Lipid Tox, indicating that DHRS3 led to an accumulation of lipid droplets (LDs). LD accumulation in the ER and cell-outgrowth change were confirmed in NH12 and SK-N-SH cells (Figure 1A(b,c)). The accumulation was observed in NH12 and SK-N-SH cells shortly after transfection (Figure 1A(a)), and then the DHRS3 expression was moved into LDs, but this phenomenon was absent in TGW cells. Electron-microscopical examination revealed the existence of small vesicles, including DHRS3 in cytoplasm, suggesting the existence of LDs (Figure 1B). CYP26A1 was also expressed in the nucleus and ER, and NROB1 was expressed in the nucleus (Figure 1A(b)). We then examined expression of the exogenous and endogenous candidate prognostic factor genes by Western blotting (Figure 1C). Expression of transfected genes in these overexpressed clones was found, except for NROB1 in TGW cells, and the molecular weights using these antibodies were matched by Western blotting. In the NROB1-transfected TGW cells, severe gene dropout and low levels of protein expression were detected during each passage of cell culture after transfection. As a result of this, it seemed to be difficult to detect the exogenous and endogenous NROB1 protein.

In clones in which siRNA for each gene was transfected, no obvious changes in protein expression were detected because the endogenous protein-expression levels of these factors in these NB cell lines were originally low. SiRNA-expressing clones showed few morphological changes, and the cells were not prone to cellular senescence or death. No change in expression levels of these genes could be detected by Western blotting with the introduction of siRNA.

In the growth assay (Figure 2A), all SK-N-SH and NH12 clones transfected with one of the three genes showed reduced growth rates. In the TGW cells, growth rates were also reduced, except for the clone transfected with DHRS3. On the other hand, the growth rates of the siRNA-expressing clones were not significantly changed (Figure 2B).

The soft-agar colony-formation assay showed that the colony sizes of DHRS3-, CYP26A1-, and NROB1-overexpressing clones were smaller than those of Mock ones (Figure 3). The colonization efficiencies were significantly reduced in the DHRS3- and NROB1-overexpressing clones of SK-N-SH cells. The numbers of small colonies formed were increased relative to Mock colonies in these cell lines, but not significantly. Colony formation assays showed that TGW cells showed no remarkable change in size and formation efficiency, while SK-N-SH and NH12 cells showed smaller colony sizes (Figure 3, Appendix A).

### 3.2. Gene-Expression Analysis

We identified the up- or down-regulated genes in each cell line after transfection of one of these three genes using RNA sequencing analysis. Genes with >2-fold down- or up-regulation compared to the control were extracted. We then carried out pathway analyses of the genes commonly changed between each cell line after gene transfection.

We found that 811 genes were more than two-fold up- or down-regulated in more than two of the three DHRS3-transfected cell lines, with 303 genes in CYP26A1-transfected cell lines and 275 genes in NROB1-transfected cell lines.

In SK-N-SH cells with DHRS3 overexpression, there were 23 genes that were more than 10-fold up-regulated and 211 genes that were more than 10-fold down-regulated. Excluding those with low reads (less than 100 read counts), we extracted 5 up-regulated genes (DHRS3, ELFN1, TAC3, SMOC1, and NME1-NME2) and 109 down-regulated genes involved in cell differentiation and cell adhesion, including LIF, CD44, COL3A1, COL5A1, THBS1, and THBS2. Pathway analysis was performed on these extracted genes with GeneMANIA, IPA, and Strand STS. At *p* < 0.05, 10 up-regulated pathways, including Vitamin A and Carotenoid Metabolism, GPCR_Ligand_Binding, and WNT_Ligand_Biogenesis_and_Trafficking, were extracted. Forty-six down-regulated pathways were extracted, including interactions related to cellular senescence and cell adhesion, such as Oncostatin_M_Signaling_Pathway, Senescence_and_Autophagy, Focal_Adhesion, and Integrin_Cell_Surface_Interactions (Appendix A).

Pathway analysis also suggested the participation of pathways related to cell adhesion and differentiation produced by overexpression of DHRS3 (Appendix A). Next, we identified transcription factors predicted to control the genetic groups in the transcription-factor-prediction tool T facts (*p* < 0.05), including JUN, SMAD3, EGR1, SMAD2, GLI1, CTNNB1, and SMAD (Appendix A). We also found that the transcription factor ESR1 controlling TAC3 was up-regulated and that it was controlled by 117 transcription factors that were down-regulated. In addition, CYP26A1-transfected cells showed changes in gene expression related to Hepatic Fibrosis and Adipogenesis, Cell Cycle, and others, including genes related to nervous system diseases, Cancer, and Developmental Disease, in NROB1-transfected cells.

### 3.3. Retro-X Tet-On Advanced System Analysis

We induced the expression of transfected genes with Dox. Gene expression started 6 h after induction, and morphological changes resembling cellular senescence (giant and markedly flattened cells) were observed after 3–5 days in DHR3- and NROB1-induced cell lines. When DHRS3 was induced by Tet-On expression, the cell-growth rate was suppressed in SK-N-SH and NH12 cell lines, but no change in the growth rate in SK-N-SH cells with CYP26A1 expression was observed, and the cell-growth rate was promoted in NH12 cells with CYP26A1 induction. When NROB1 was induced by Tet-On expression, the cellular growth rate of NH12 cells was suppressed (Figure 4, Appendix A).

These induced genes were widely distributed within the cytoplasm of NB cell lines, but after more than 5 days of DHRS3 induction in SK-N-SH and NH12 cells, DHRS3 translocated to the membrane surface of LDs, similar to what was seen in the constitutively expressed plasmids (Appendix A). NH12 and SK-N-SH cell lines showed cell death more than 6 days after induced expression of DHRS3, and growth was no longer detected when they were cultured for more than 10 days. When gap-closure assays were performed to examine cellular migration, DHRS3 or NROB1 induction in SK-N-SH and NH12 cell lines slowed down-regulation to close the gap, while SK-N-SH and NH12 cells with CYP26A1 induction remained unchanged (Appendix A).

Cell-cycle analysis of the Dox-induced expression of DHRS3 of SK-N-SH cells suggested that G0–G1 arrest occurred (Figure 4e). There was no change in cell-growth rate or cell cycle in CYP26A1-induced cells. Cell-growth rates were suppressed in NROB1-induced NH12 cells, but this expression did not affect the cell cycle. The cell sizes after tis expression became larger than before Dox induction (Appendix A). Cellular senescence assays showed that NROB1-induced cell lines expressed senescence markers and underwent cellular senescence, whereas DHRS3-induced cells expressed fewer senescence markers.

Next, in the differentiation-inducing experiment with 1–10 µM of ATRA, SK-N-SH and NH12 cells with DHRS3 expression by Dox induction showed flattened and giant morphologies and more strongly expressed senescence markers compared to cells without ATRA (Figure 5). At this time, there was little change in the expression of p16^INK4a^, a senescence-associated marker, and the expression of p21Waf1/Cip1 was enhanced, suggesting that DHRS3 may be involved in senescence and differentiation.

## 4. Discussion

The prognosis of patients with NB depends on the tumor biology and the stage of disease [10,11,12]. To better understand NB biology, many studies have been conducted and several factors related to highly malignant NB have been identified, including *MYCN* amplification and telomerase activation [13,14,15,16,17]. However, few basic studies have assessed which factors characterize favorable tumors. Therefore, we conducted the present study, in which we have shown that *DHRS3*, *CYP26A1*, and *NROB1* are potential favorable prognostic factors for NB, as the expression of these genes in NB cell lines suppressed growth and reduced anchorage independency in plasmid-established lines.

In the cells transfected with one of these genes, morphological changes were largely similar in these NB cell lines, and our findings suggest that overexpression of these genes induces cellular senescence. Morphologically, senescence-like changes, such as enlargement and flattening of cells, were observed in the clones after transfection. In the growth assay, overexpression of these genes reduced growth in most cell lines. Finally, soft-agar formation suggested that overexpression of these genes reduced growth and cellular differentiation via a lower level of stemness activity.

Transfected DHRS3 accumulated in the ER and LDs in NH12 and SK-N-SH cells but was absent in TGW cells. In the growth assay (Figure 4A), DHRS3-transfected SK-N-SH and NH12 cells showed reductions in growth, but growth was not changed in TGW cells. In the soft-agar colony-formation assay, TGWs showed little change in size and formation efficiency. Since TP53 promotes the accumulation of DHRS3 in LD and ER [18], this was not seen in TGWs harboring a mutated TP53 gene. The TP53 mutation might thus be one of the genetic aberrations that are characteristic of highly malignant NB, and TP53 is necessary for this growth-reduction pathway through DHRS3. On the other hand, NROB1-transfected TGW cells showed severe gene dropout and low levels of protein expression, indicating the existence of a TP53-independent pathway of cellular senescence by NROB1.

Gene expression analysis by RNA sequencing revealed that pathways related to cellular senescence and cell adhesion were related. Pathway analysis of significantly up- and down-regulated genes found a reduction in several pathways, including Vitamin A and Carotenoid Metabolism, GPCR Ligand Binding, and WNT Ligand Biogenesis and Trafficking, implicating cellular senescence, cell adhesion and differentiation, and nervous system development.

DHRS3 is a member of the classical short-chain dehydrogenase reductase superfamily [19], which was originally cloned from retinal cells. Retinol-like compounds affect intracellular processes mainly by regulating gene expression, including cell proliferation and differentiation [20]. They are expressed in several non-ocular human tissues, including the liver, pancreas, heart, kidney, and lung. Although DHRS3 was found to be up-regulated in papillary thyroid cancer and neuroblasts, DHRS3 was negatively correlated with the metastasis of papillary thyroid carcinoma and was associated with a better prognosis for NB [3,21]. This is consistent with the data for DHRS3 overexpression in our study. Nonetheless, further research is necessary to understand the specific role of DHRS3 in NB. Notably, in a sarcoma study, it was found that a four-gene signature, including DHRS3, JRK, TARDBP, and TTC3, can predict patient survival based on a real-world cohort. These genes may be more feasible for clinical application and should be validated prospectively.

NROB1 (also known as DAX1) is a critical mediator of retinoic-acid (RA)-induced nNOS gene transcription in NB [5]. Retinoids induce CYP26 enzyme production in the liver, enhancing their own rapid metabolic clearance, while retinoid resistance in tumor cells occurs in part due to increased CYP26 production. RA-metabolism-blocking agents, which inhibit CYP26 enzymes, can improve RA pharmacokinetics in preclinical NB models [22]. The signaling networks through which RA mediates NB differentiation were inhibited upon *MYCN* overexpression, which revealed opposing regulation of RA and for a number of differentiation-relevant genes, including LMO4, CYP26A1, ASCL1, RET, FZD7, and DKK1 [23]. Furthermore, CYP26A1 was found to be a retinoid-responsive gene in retinoid-sensitive NB in *MYCN* transgenic mice and lung and breast cancer cell lines [24].

Based on the cellular phenomena and gene expression profiles we observed in the induction experiments on transfected NB cell lines, we conducted a differentiation-inducing experiment by administering ATRA to transfected cells. SK-N-SH and NH12 cells with DHRS3 expression became flattened and exhibited a giant morphology, and they strongly expressed senescence markers compared to cells without ATRA exposure. We observed little change in the expression of p16^INK4a^, a senescence-associated marker, and enhanced expression of p21Waf1/Cip1, suggesting that DHRS3 may be involved in senescence and differentiation.

Dropout of the transduced genes occurred during culture passage in transfected NB cell lines, making long-term stable gene transfer difficult. Thus, having a tetracycline-inducible expression system with retroviral vectors resulted in more pronounced cell-morphology changes. Though the expression of CYP26A1 did not affect the cell-growth rate or cell cycle in the Dox-expressing system, DHRS3 and NROB1 expression led to cell-morphology changes and inhibition and arrest of cell growth in a short period of time. Notably, DHRS3 expression arrested the cell cycle by engaging with the all-trans-retinol pathway and undergoing differentiation and senescence. Since p53 promotes LD accumulation in a manner consistent with DHRS3 enrichment in the ER [18], TGW cells were resistant to DHRS3 induction due to the TP53 gene mutation they harbor. On the other hand, CYP26A1 induction in the Dox-expressing system showed morphological changes without growth inhibition (Figure 4, Appendix A), suggesting that CYP26A1, as a retinoid-inducible enzyme, might play a role as a cell-differentiation-inducing factor instead of inhibiting growth arrest—a matter which requires further analysis.

Candidate genes for favorable prognostic factors may reduce the ability of NB cell lines to suppress tumor growth and anchorage dependency. To examine the usefulness of these findings as diagnostic markers of malignancy, we will next examine clinical samples from a bank of intractable tumors. These findings will offer insights into molecular targets and the feasibility of differentiation-induction therapies for the treatment of NB.

## Figures and Tables

**Figure 1 cells-11-03171-f001:**
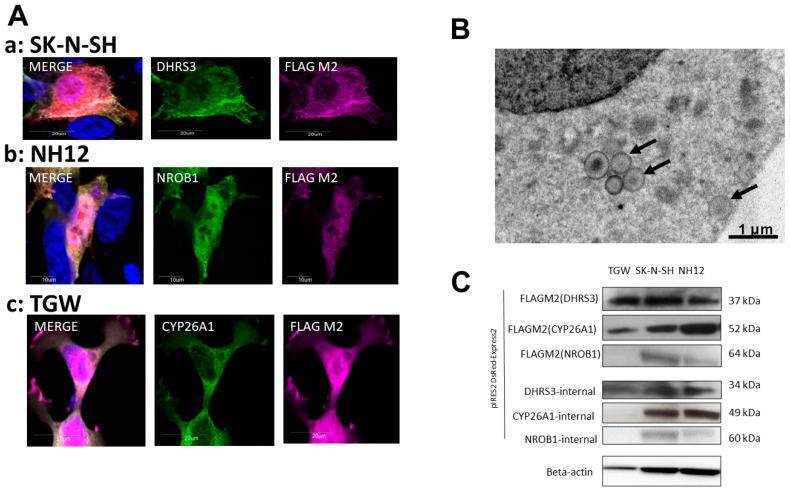
Transfection efficiency in SK-N-SH, NH12, and TGW neuroblastoma cell lines. Representative images of transfected, undifferentiated neuroblastoma cell lines. (**A**) Stable expression levels of transfected gene products were assessed with fluorescent immunostaining in transfected cell lines (a–c). Expression of exogenous proteins was detected with FLAG M2 antibodies and their respective endogenous antibodies. In established cell lines, DHRS3 was expressed mainly in the plasma membrane and endoplasmic reticulum (a,b). The merged photos include nuclear staining by DAPI. (**B**) DHRS3-transfected strains were lipid-stained using Lipid Tox and visualized by transmission electron microscopy, and we found an accumulation of DHRS3 in small vesicles (arrows). These vesicles were observed shortly after transfection in NH12 and SK-N-SH cells but were absent in TGWs. In addition, CYP26A1 was also expressed in both the nucleus and the endoplasmic reticulum, and NROB1 was expressed in the nucleus (b). (**C**) The presence of the exogenous proteins was confirmed by their expected molecular weights by Western blotting.

**Figure 2 cells-11-03171-f002:**
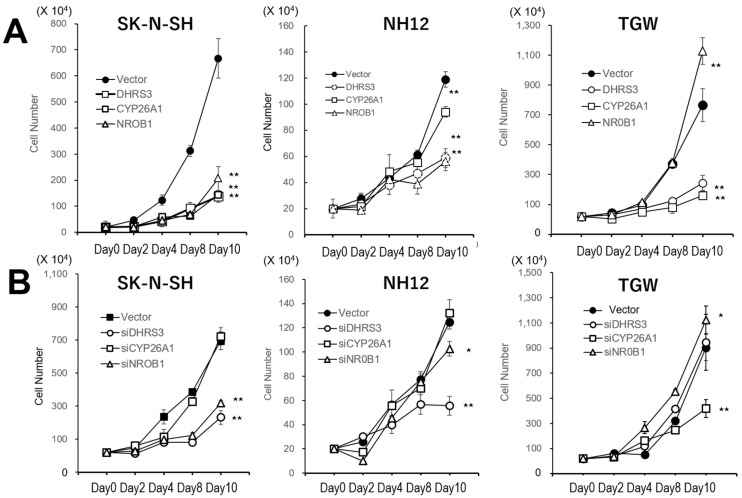
Cell-cycle analysis of cells by gene transfection or siRNA transduction. (**A**) Growth curves of neuroblastoma cells transfected with one of these genes selected by the favorable tumors. Overexpression of these factors reduced the growth rates. (**B**) Growth rates of siRNA transduction for each gene in these cell lines. The growth rates were not affected by siRNA transduction, but some siRNA transduction reduced the growth rates. Data points represent means ± SDs of triplicate wells. The growth rates of transfected cells were reduced, except for DHRS-transfected TGW cells. Native cell lines were confluent at 4–5 days in this culture condition. **: *p* < 0.01, *: *p* < 0.05.

**Figure 3 cells-11-03171-f003:**
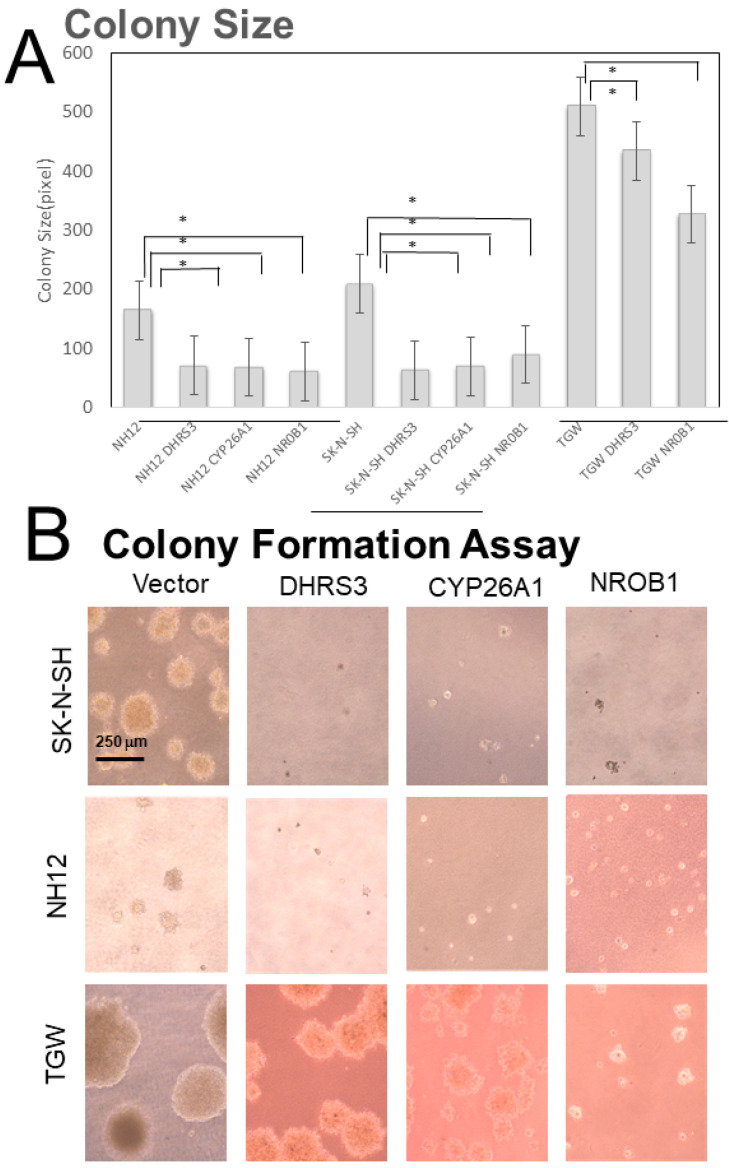
Soft-agar colony-formation assay. (**A**) The colony sizes in DHRS3-, CYP26A1-, and NROB1-overexpressing clones were significantly reduced (*p* < 0.05). The CYP26A1-overexpressing TGW cells were hard to obtain, so statistical evaluation was not performed. *: *p* < 0.05 (**B**) Cell morphology. The sizes of colonies were obviously large in TGW cells.

**Figure 4 cells-11-03171-f004:**
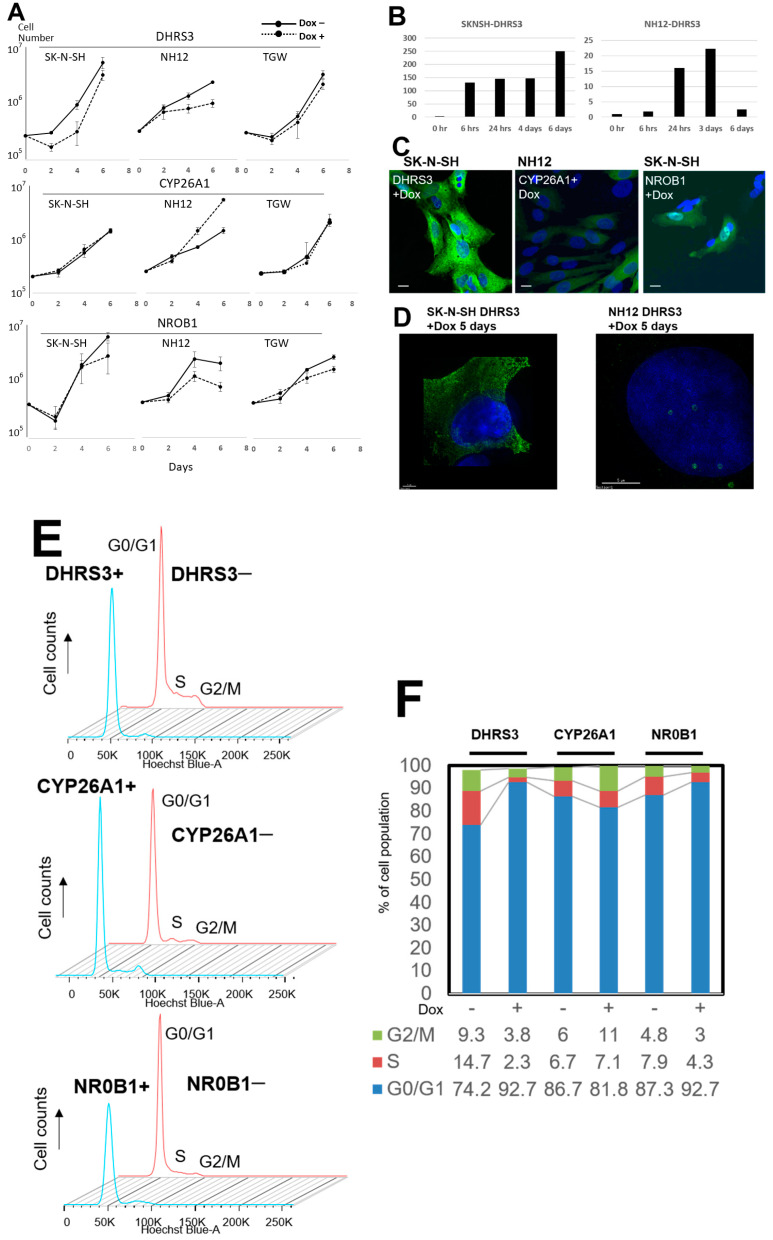
Cell-growth curves of NB cells induced by the Retro-X Tet-On Advanced System. (**A**) Growth curves of NB cell lines. Gene expression was induced with or without doxycycline. (**B**) Time course of the *DHRS3*-gene-expression analysis in SK-N-SH and NH12 cells using qPCR. Gene expression of DHRS3 was up-regulated until 3–6 days after Dox induction. (**C**) Immunofluorescent imaging of gene expression using the mouse-specific FLAG-M2 monoclonal antibody. Expression of each gene was obvious in all cell lines 5 days after Dox induction, but expression of DHRS3 decreased at 7 and 9 days after Dox induction. Scale bar is 5 µm. (**D**) Morphological alterations of SK-N-SH and NH-12 cells at 5 days after Dox induction. SK-N-SH and NH12 cells showed cellular enlargement. Expression of DsRed corresponded to inserted gene expression. (**E**) Cell-cycle analysis using flow cytometry in SK-N-SH cells; G0/G1 arrest was detected in Dox-induced DHRS3-expressing (DHRS3+) and NROB1-expressing (NROB1+) cells, but not in CYP26A1-expressing (CYP26A1+) cells. (**F**) Percentages of cells in G0/G1, S, and G2/M for the cells with or without Dox induction of each gene. Reductions in the percentages of S and G2/M were detected in the SK-N-SH DHRS3+ and NROB1+ cells, but not in CYP26A1+ cells.

**Figure 5 cells-11-03171-f005:**
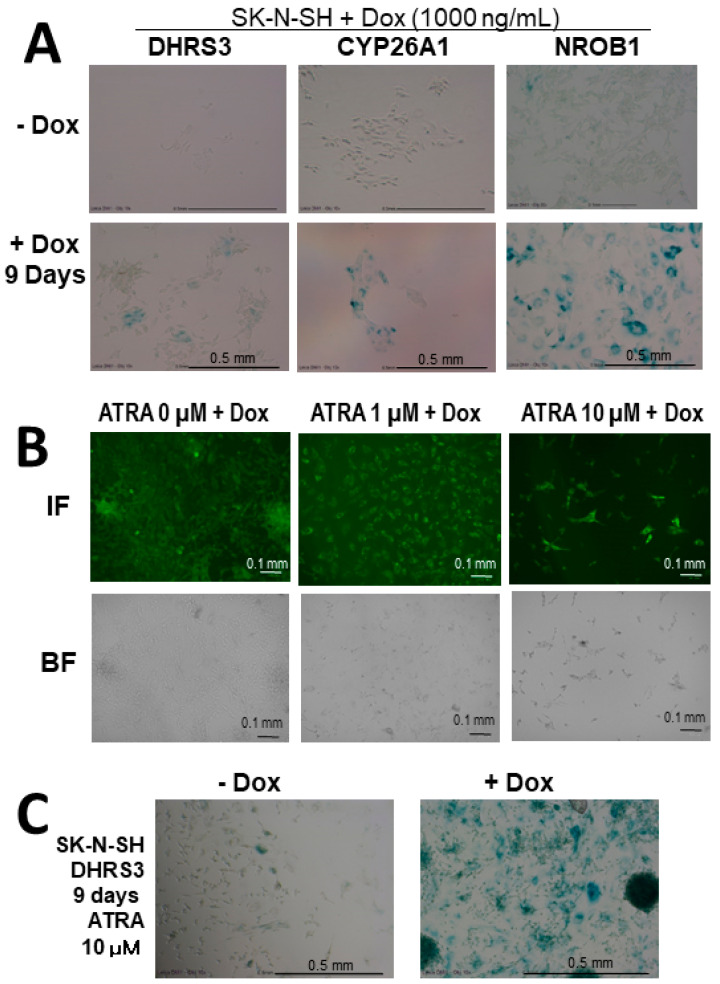
ATRA-inducing differentiation experiments in Dox-treated NB cells. (**A**) Expression of senescence markers in Dox-treated SK-N-SH NB cell lines. SA-β-Gal-positive cells (green). SK-N-SH cells were untreated (−Dox) or Dox-treated until day 9 (+Dox). Ectopic Dox-induced SK-N-SH cells were observed to express high levels of a senescence marker. (**B**) Expression of Dox-treated DHRS3-expressing SK-N-SH NB cells with or without ATRA treatment. After ATRA treatment, SK-N-SH cells were assessed by immunofluorescence. High-dose ATRA induced cellular enlargement, cell death, and strong expression of DHRS3. (**C**) Expression of a senescence marker in SK-N-SH cell lines after 10 μM of ATRA treatment. SA-β-Gal-positive cells (green). The senescence marker was higher in the Dox-treated SK-N-SH cells.

## Data Availability

The multiplex qPCR data analyzed in this study are included in the article and in the Appendix A. The sequencing data analyzed in this study are not publicly available but can be obtained from the corresponding author. NGS sequencing data have been uploaded into DDBJ (DNA Data Bank of Japan) with the accession number PRJDB14234.

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
