# Peer review of "In Vitro Transfection of Up-Regulated Genes Identified in Favorable-Outcome Neuroblastoma into Cell Lines"

_cells, 2022, doi:10.3390/cells11193171_

Round 1

Reviewer 1 Report

Reviewer's comments:

Fig 1 L 203.  repeat Lipid droplets (lipid droplet) ...?

  Part of a cell is shown with uktrastructure, why...lipid droplets should be marked.

pages 2-4: Material and Methods:

 there is no electron microscopy technique indicated in this section

The reviewer has no access to the supplementary figures

Author Response

Fig 1 L 203.  repeat Lipid droplets (lipid droplet) ...?

  Part of a cell is shown with uktrastructure, why...lipid droplets should be marked.

A: Thank you for this comment. We revised Lipid droplets (lipid droplet) to “small vesicles (arrows)” and added the marks for indicating them.(lines 225-6)

pages 2-4: Material and Methods:

 there is no electron microscopy technique indicated in this section

A: I added the electron microscopic procedure in methods (lines 151-159).

The reviewer has no access to the supplementary figures

A: The supplementary file will be resent to you.

Reviewer 2 Report

Very interesting concept of taking genes that are upregulated in good prognosis neuroblastomas and transfecting them into cell lines. A few points regarding the methods and data. It would be important to further characterize the cell lines that you used for your experiments. Would they themselves be considered favorable or unfavorable? For example do the express MYCN or other poor prognostic markers. Second how do you explain the lack of protein expression in the TGW cell lines after transfection of CYP26A1 and NROB1? And is this why you do not see decreased growth rates in colony formation in this cell line? Although the concept is interesting, I feel you are far from any therapeutic potential but it is a good first step. I am however, concerned with the conflicting results you obtained in the Retro-X-Tet transfection. In the results you mention that there was no change in growth rate in some of the cell lines with some of the transfections and in fact there was increased growth in the NH12 cells with CYP26A1 induction. I think it would be important to discuss this further in the discussion. The main issue I had with the paper was the myriad grammatical errors. They are too numerous to point out here but one example would be the sentence in line 51-52 that starts: To rule out… we also the retroviral… There is no verb in that sentence. I assume you meant we also used the retroviral or something such.

Author Response

Very interesting concept of taking genes that are upregulated in good prognosis neuroblastomas and transfecting them into cell lines. A few points regarding the methods and data. It would be important to further characterize the cell lines that you used for your experiments.

Would they themselves be considered favorable or unfavorable? For example do the express MYCN or other poor prognostic markers.

A: All neuroblastoma cell lines were derived from unfavorable tumors because no cell lines have established from favorable neuroblastoma. Our experiment chose the following three different types cell lines: SK-N-SH were derived from INSS 4 tumor of 4 years old girl with chromosomes 9q+ and 22q+., Chromosomes N7 is trisomic and a single MYCN copy.. NH-12 was established from our case with INSS 4 MYCN amplified adrenal tumor of 15-month old boy. TGW was derived from a 23-month old Japanese boy with disseminated neuroblastoma has MYCN amplification and TP53 mutation. We used these difference character cell lines derived from unfavorable neuroblastoma.

Second how do you explain the lack of protein expression in the TGW cell lines after transfection of CYP26A1 and NROB1?

A: The transfection of the vector into TGW cells was so difficult that the expression levels of these genes were considered to be undetectable. However, the slight upregulation of these genes were confirmed using NGS and RT-PCR. This explanation was described in lines 238-240.

And is this why you do not see decreased growth rates in colony formation in this cell line?

A: We did not measure the growth rates in colony formation assay. We usually measured the number and sizes of colonies after 48 hrs.

 Although the concept is interesting, I feel you are far from any therapeutic potential but it is a good first step. I am however, concerned with the conflicting results you obtained in the Retro-X-Tet transfection. In the results you mention that there was no change in growth rate in some of the cell lines with some of the transfections and in fact there was increased growth in the NH12 cells with CYP26A1 induction. I think it would be important to discuss this further in the discussion.

A: Thank you for the important comment. According to this suggestion, we added the discussion for this conflicting result in lines 440-443.

The main issue I had with the paper was the myriad grammatical errors. They are too numerous to point out here but one example would be the sentence in line 51-52 that starts: To rule out… we also the retroviral… There is no verb in that sentence. I assume you meant we also used the retroviral or something such.

A: Thank you for your comment. This paper was reviewed by the native-English speakers and revised in these grammatical errors.

Reviewer 3 Report

This study investigates the effect of ectopic expression of NHRS3, NROB1 and CYP26A1 on neuroblastoma cell lines. The rationale for selecting these three genes and three cell lines is not clearly presented. Were these genes differentially expressed in favorable tumor in comparison to unfavorable ones? Overall, the experimental results presented in the manuscript do not adequately support the conclusions. While it is interesting that “overexpression of these genes induces senescence”, this statement was however not substantiated with direct and quantitative analyses of cellular senescence and the relationship between gene expression levels and cellular growth arrest. Moreover, the growth arrest effect appears to be NB cell line-dependent and gene-dependent. Furthermore, it is puzzling why both overexpression and siRNA knockdown of DHRS3 suppress cell growth in SK-N-SH and NH12 cell line. Below are some of specific comments:  

-          Methods for cell transfection and analysis are lacking.

-          No data presented for the statement line 192-193 (Result 3.1). Was this senescence statement based on morphology only?

-          Fig. 1:  there were no controls. “Stable expression” (line 198): dose is mean certain number of cell passage? Or it just meant “shortly after transfection”? A better term could be used. The legend needs clarification for experimental conditions.

-          Fig. 1A: From the representative images, it is difficult to see that “NROB1 was expressed in the nucleus”.  Fig. 1B: the vesicles do not appear to be lipid droplet (LD) on the TEM. There is no method section on TEM. If DHRS3 overexpression increases LD formation, this could be readily examined.

-          Fig. 2: cell death as a potential contributing factor for decreased cell number over time needs to be considered. If the protein expression in siRNA experiments was not different from controls, why would knockdown DHRS3 suppress cell growth?

-          Fig 3: need clarification for mock treatment. No scale bars (Figs. 3 and 4). Fig. 3 legend mentioned that CYP26A1-overexpressing TGW cells could not be obtained (thus no data for colony size in Fig. 3A.), however 3B showed CYP26A1 TGW colony.

-          Fig. 4 & 5A appear to demonstrate that Dox-induced expression of DHRS3 and NROB1 induces growth arrest and morphological changes resembling that of senescence. However, there is no statistical analysis of cell growth (Fig. 4A), no direct results to demonstrate the statement on NROB1-induced senescence (line 321-322). Was cell growth analysis based on total cells or viable cells as time-dependent cell death induced by gene expression may complicate data interpretation? In Fig. 4E, DOX++ group (higher specific gene expression or higher dose of Dox treatment?) appeared to have much less cells than Dox negative controls. But Fig. 4A did not show a significant drop in cell number at day 6.

-          One general concern is image quality. Many cell culture images are of poor quality for visualizing cell morphology, making it difficult to interpretate the results (e.g., Fig. 5B). Some experiments lacked proper controls (e.g., Fig. 5B, no corresponding ATRA-Dox controls). Quantitative analysis for cell migration and beta-gal+ population is also missing.

-          In addition, there are a few places that figure and text are mismatched. For example, line 312 Fig. S4d, yet there is no d in Fig. S4.

-          The manuscript contains multiple typos and errors. For example: line 49-50 & line 53-53 (confusing sentence & unfinished sentence), lines 53-54, 90, 232, Figure labels (e.g., Fig. 1A, Fig. 5A), NR0B1 or NROB1, etc.  

Author Response

This study investigates the effect of ectopic expression of NHRS3, NROB1 and CYP26A1 on neuroblastoma cell lines. The rationale for selecting these three genes and three cell lines is not clearly presented. Were these genes differentially expressed in favorable tumor in comparison to unfavorable ones?

A: Yes. This explanation was added in the introduction (lines 36-38)  

Overall, the experimental results presented in the manuscript do not adequately support the conclusions. While it is interesting that “overexpression of these genes induces senescence”, this statement was however not substantiated with direct and quantitative analyses of cellular senescence and the relationship between gene expression levels and cellular growth arrest.

A: Thank you for your comments

Moreover, the growth arrest effect appears to be NB cell line-dependent and gene-dependent. Furthermore, it is puzzling why both overexpression and siRNA knockdown of DHRS3 suppress cell growth in SK-N-SH and NH12 cell line. Below are some of specific comments:  

A: As mentioned in this manuscript, the original expression levels of these genes were very low or null. The growth inhibition effect of SiRNA expression might be derived from siRNA transfection or non-target effects of siRNA.

-          Methods for cell transfection and analysis are lacking.

A: According to this suggestion, we added the method of transfection and cell analysis (lines 565-573)

The day before transfection, culture cells were trypsinized and counted. Seeded cells in 24-well plate to make 50~70% confluent on the day of transfection in 100 ul of Serum free Opti-MEM® I Reduced Serum Medium with 500 ng of plasmid DNA and then 0.5 ul of PLUS Reagent was added. After incubation for 5 min at room temperature (r.t.), the cells were mixed throughtly with Lipofectamine LTX and then incubated for 30 min at r.t. These cells were divided into each well and incubated for 6 hrs at 37℃ with 5% CO2. Selection process was started from the next day for analyzing the concentration determined in the preliminary experiment.

-          No data presented for the statement line 192-193 (Result 3.1). Was this senescence statement based on morphology only?

A: Yes, we evaluated senescence based on morphology. As using Retroviral constitutive transfection, the morphology of cells changes remarkably flattened as shown in Suppl. Fig.4.

On the other hand, such changes could not be found in the cells with vector transfection.

-          Fig. 1:  there were no controls. “Stable expression” (line 198): dose is mean certain number of cell passage? Or it just meant “shortly after transfection”? A better term could be used. The legend needs clarification for experimental conditions.

A; We measured the expression levels of control (cells without transfection) using Western blotting. This data was added in supplementary Fig. “Stable expression” means the cells with stable overexpression of transfected gene after certain number of cell passage. Therefore, it did not mean “shortly after transfection”.

According to this suggestion, figure legends were revised. (line 220)

-          Fig. 1A: From the representative images, it is difficult to see that “NROB1 was expressed in the nucleus”.  Fig. 1B: the vesicles do not appear to be lipid droplet (LD) on the TEM. There is no method section on TEM. If DHRS3 overexpression increases LD formation, this could be readily examined.

A: As mentioned by this comment, it is difficult to be observed in the localization of nuclei. It is natural that NROB1 was located at nuclei and cytoplasm. As mentioned by this comment, these vesicles were not confirmed as “lipid droplet (LD)” because there were no data to clarify these vesicles as LD. Therefore, we revised this to “small vesicles”. We detected the localization of DHRS3 in these vesicles.

A: According to this comment, we added methodology of TEM and LD like vesicle with DHRS3 will be readily examined.

-          Fig. 2: cell death as a potential contributing factor for decreased cell number over time needs to be considered. If the protein expression in siRNA experiments was not different from controls, why would knockdown DHRS3 suppress cell growth?

A: This comment is correct. As mentioned in this manuscript, the original expression levels of these genes were very low or null. Therefore, the protein expression in siRNA experiments was not different from controls and the growth inhibition effect of siRNA expression might be derived from siRNA transfection or non-target effects of siRNA.

-          Fig 3: need clarification for mock treatment. No scale bars (Figs. 3 and 4). Fig. 3 legend mentioned that CYP26A1-overexpressing TGW cells could not be obtained (thus no data for colony size in Fig. 3A.), however 3B showed CYP26A1 TGW colony.

A: According this comment, we added scale bars in these figures,

We could not obtain the sufficient TGW cells which overexpress CYP26A1after transfection because of difficulty of the passage. However, we got certain amount of the CYP26A1-overexpressing TGW cells for analyzing colony formation assay.

-          Fig. 4 & 5A appear to demonstrate that Dox-induced expression of DHRS3 and NROB1 induces growth arrest and morphological changes resembling that of senescence. However, there is no statistical analysis of cell growth (Fig. 4A), no direct results to demonstrate the statement on NROB1-induced senescence (line 321-322).

A; This comment is important. We added the explanation of this result in discussion (lines 439-442),

Was cell growth analysis based on total cells or viable cells as time-dependent cell death induced by gene expression may complicate data interpretation?

A; The growth analysis was used the total cells including floating cells. Therefore, the number of viable cells might be different. These were added line 65.

In Fig. 4E, DOX++ group (higher specific gene expression or higher dose of Dox treatment?) appeared to have much less cells than Dox negative controls. But Fig. 4A did not show a significant drop in cell number at day 6.

A: DOX + and ++ means the expression levels of Dox induced gene. As mentioned by this comments, it was hard to understand. Therefore, this Fig. 5 was changed to more easy to understand.

-          One general concern is image quality. Many cell culture images are of poor quality for visualizing cell morphology, making it difficult to interpretate the results (e.g., Fig. 5B). Some experiments lacked proper controls (e.g., Fig. 5B, no corresponding ATRA-Dox controls). Quantitative analysis for cell migration and beta-gal+ population is also missing.

A: The images were changed to those with high quality, as possible as we could.

-          In addition, there are a few places that figure and text are mismatched. For example, line 312 Fig. S4d, yet there is no d in Fig. S4.

-          The manuscript contains multiple typos and errors. For example: line 49-50 & line 53-53 (confusing sentence & unfinished sentence), lines 53-54, 90, 232, Figure labels (e.g., Fig. 1A, Fig. 5A), NR0B1 or NROB1, etc.  

A: Thank you for this comment. We notice several typo grammatical errors. We corrected them. And since we are not English native speakers, this paper is now under checking by native speakers.